# Impacts of non-ideality and the thermodynamic pressure work term p∆v on the Surface Energy Balance

William J. Massman[1]

[1]USDA Forest Service, Rocky Mountain Research Station, 240 West Prospect Road, Fort Collins, CO 80526 USA

**Correspondence:** W. J. Massman (wmassman@fs.fed.us)

**Abstract.** Present day eddy covariance based methods for measuring the energy and mass exchange between the earth's surface and the atmosphere often do not close the surface energy balance. Frequently the turbulent energy fluxes (sum of sensible and latent heat) underestimate the available energy (net incoming radiation minus the soil conductive heat flux) by 10 to 20% or more. Over the last three or four decades several reasons for this underestimation have been proposed, but nothing completely definitive has been found. This study examines the contribution of two rarely discussed aspects of atmospheric thermodynamics to this underestimation: the non-ideality of atmospheric gases and the significance the water vapor flux has on the sensible heat flux, an issue related to the pressure work term $p\Delta v$. The results were not unexpected, i.e., these effects are too small to account for all of the imbalance between the sum of the turbulent fluxes and the available energy. Together they may contribute 1-3% of the difference (or 10 to 15% of the percentage imbalance).

## 1 Introduction

The microclimate at any given location on the earth's surface is determined by a balance between the incoming and outgoing energy. Documenting and measuring these energy flows is fundamental to micrometeorology and to the understanding of the functioning of the earth's ecosystems (e.g., Geiger et al. , 2003). In its simplest form the surface energy balance (SEB) is composed of four terms: $R_n = L_v E + H + G$; where $R_n$ (Wm$^{-2}$) is net radiation (= incoming radiation minus reflected and outgoing infrared radiation), $L_v E$ (Wm$^{-2}$) is the latent heat flux or the energy required to evaporate (and transpire) moisture, $H$ (Wm$^{-2}$) is the sensible heat flux associated with heated air currents as they move upward and away from the surface, and $G$ (Wm$^{-2}$) is the heat conducted into the components of the surface (soil, tree branches and trunks, ...). For the purposes of the present study all other terms of the SEB, which tend to be small, can be ignored. But despite decades of effort micrometeorologists worldwide have not been able to achieve a fully satisfactory level of closure to the SEB (e.g., Twine et al. , 2000; Oncley et al. , 2007; Leuning et al. , 2012).

There have been many studies that have proposed explanations for the often observed imbalance, but the present study focuses on only two, (Paw U et al. , 2000, Appendix C) and Kowalski (2018), which are centered exclusively on $L_v E$ and

$H$. The authors of both of these studies seek at least a partial "solution" to the energy imbalance problem by suggesting that the pressure work term, $p\Delta v$ (Jkg$^{-1}$), that part of the first law of thermodynamics that accounts for the work done on a system or by a system during the physical expansion or compression of that system, has not been incorporated correctly into micrometeorological theory underpinning the measurements of $L_v E$ and $H$. Kowalski (2018) argued (incorrectly) that the enthalpy of vaporization, $L_v$ (Jkg$^{-1}$), did not include $p\Delta v$. So he proposed adding $p\Delta v$, which by his analysis was equal to

the term $R_d T_v$ (where $R_d$ is the specific gas constand for dry air (Jkg$^{-1}$K$^{-1}$) and $T_v$ is the virtual temperature of the air (K)), to correct $L_v$, yielding in turn a 3-4% increase in $L_v$. But, as pointed out by the reviewers and commenters on Kowalski's study, adding $p\Delta v$ to $L_v$ is incorrect because $p\Delta v$ is by defintion a component of $L_v$ and so adding it to $L_v$ would be double counting it. Furthermore, as noted by another commenter, $p\Delta v$ can be computed directly for the evaporative process and it is not equal to, nor numerically the same as, $R_d T_v$ (also see Figure 1 and related discussion below).

Paw U et al. (2000, Appendix C), on the other hand, take a different approach to the $p\Delta v$ term. They do not apply their correction directly to $L_v E$ in the SEB equation. Rather they apply their correction to the heat flux, $H$, based on a change in density of an air parcel associated with mixing newly transpired or evaporated water vapor with the air contained within that air parcel. They pose their correction in terms of an equivalent temperature perturbation, such that after evaporation has occurred the (turbulent + diffusive) transport-driven expansion of the water vapor into the atmosphere surrounding the source of water

vapor (e.g., plant stomatal pores and the porous soil) results in a change in the atmospheric density that is associated with a concomitant change in the atmospheric temperature. So in effect Paw U et al. (2000) are using the first law of thermodynamics (expressed in terms of atmospheric processes and the pressure work term) to argue that $H$ should be adjusted to include a small term that is proportional to the mass flux of water vapor, $E$ (kgm$^{-2}$s$^{-1}$).

The present paper employs "classical" thermodynamics to examine (a) the influence that the non-ideality of atmospheric

gases can have on the SEB and (b) the methods and conclusions of Paw U et al. (2000, Appendix C) regarding the first law of thermodynamics and the pressure work term's influence on the turbulent heat flux and ultimately the SEB as well. Although it is true that what I develop herein is not necessarily "new" science, some of the theory I employ may well be new to the general environmental and geo-biophysical communities. The present study is divided into two parts. The first examines and quantifies how mixing of air and water vapor as non-ideal (or real) gases, rather than as ideal gases, can have on $L_v$ and the specific heat

of moist air. In the second part the first law of thermodynamics is employed to derive the influence water vapor has on potential temperature, which in turn gives rise to an expression, different from that developed by Paw U et al. (2000, Appendix C), relating how the kinematic heat flux is influenced by the mass flux of water vapor, $E$. In summary, this study shows that any potential corrections to the SEB from either of these two sources are likely to be negligible and certainly much smaller than either Kowalski (2018) or Paw U et al. (2000) propose.

## 2  Non-ideal gases

The next three sections are a purely theoretical argument intended to estimate the influence that a mixture of non-ideal gases (water vapor and dry air) can have on the SEB near Standard Pressure and Temperature (STP) by comparing the enthalpy of vaporization of water and the specific heat of moist air associated with ideal gases and non-ideal gases. Here "near STP" will be understood as pressures between about 70 kPa and 105 kPa and temperatures between about 0 C and 100 C or so – or an atmospheric state typical of near-surface conditions on earth.

### 2.1  Enthalpy of Vaporization

The enthalpy of vaporization for pure water into an atmosphere of pure water vapor, see either Wagner and Pruß (2002) or Harvey and Friend  (2004), is expressed as

$$L_v^* = h_v^* - h_l^* \tag{1}$$

where $L_v^*$ ($\mathrm{Jkg}^{-1}$ or $\mathrm{Jmol}^{-1}$) is the enthalpy of vaporization for pure liquid water into an atmosphere of pure saturated water vapor, $h_v^*$ ($\mathrm{Jkg}^{-1}$ or $\mathrm{Jmol}^{-1}$) is the specific enthalpy of saturated vapor, and $h_l^*$ ($\mathrm{Jkg}^{-1}$ or $\mathrm{Jmol}^{-1}$) is the specific enthalpy of pure liquid water. Note (a) that the asterisk superscript ($^*$) will be used to denote a pure quantity (as opposed to a mixture which will not be superscripted) and (b) that the researchers cited above essentially employ the Clausius-Clapeyron equation to determine $h_v^* - h_l^*$. Of course, pure liquid water under near-earth-surface conditions will not be composed solely of pure liquid water. Rather it will be a mixture of pure liquid water and, e.g., dissolved atmospheric gases ($O_2$, $CO_2$, $CH_4$, etc) and possibility any number of dissolved organic and inorganic compounds (e.g., mineral salts, organic acids, etc). But for the present study, it is unnecessary of consider this additional complexity. Figure 1 includes plots of $L_v^*$ as a function of temperature, $T_K$ (degrees K), computed using the formulations of Wagner and Pruß (2002) (red line) and a linear approximation to it (black line) over the plotted temperature range.

Also included in Figure 1 are the two components of the enthalpy of vaporization (i.e., the change in internal energy, $du^*$, and the pressure work term, $p^* \Delta v^*$), where accordingly $L_v^* = du^* + p^* \Delta v^*$. For this figure $du^*$ is calculated as the difference $L_v^* - p^* \Delta v^*$ and $p^* \Delta v^*$ is estimated as follows:

$$p^* \Delta v^* = p_v \left( \frac{1}{\rho_v} - \frac{1}{\rho_l} \right) \tag{2}$$

where $p_v$ (Pa) is the vapor pressure and $\rho_v$ ($\mathrm{kgm}^{-3}$) is the vapor density and $\rho_l$ ($\mathrm{kgm}^{-3}$) is the density of liquid water. The numerical algorithms used for $p_v$, $\rho_v$ and $\rho_l$ are from Wagner and Pruß (2002). But since $\rho_l >> \rho_v$ it follows that $p_v(1/\rho_v - 1/\rho_l) \approx p_v/\rho_v$. In turn the ideal gas law yields $p_v/\rho_v = RT_K/M_w$ – also shown on Figure 1 – where $R$ ($\mathrm{Jmol}^{-1}\mathrm{K}^{-1}$) is the universal gas constant and $M_w$ ($\mathrm{kgmol}^{-1}$) is the molecular mass of water. The three quantities, $du^*$, $p^* \Delta v^*$, and $RT_K/M_w$ are included in Figure 1 primarily for the sake of completeness and to give some sense of their relative contributions to $L_v^*$. Figure 1 indicates that $p^* \Delta v^* \approx L_v^*/15$, meaning that $p^* \Delta v^*$ is a relatively small component of $L_v^*$.

Next, consider a system of $N_d$ mols of dry air and $N_l$ mols of pure liquid water separated from one another by an imperme-able membrane. Both are at the same temperature $T_{K,init}$ and the pressure of the dry air is $p_d$ (Pa). Further assume that this dry air/liquid water system is isolated, i.e., it cannot exchange mass or energy or interact mechanically with its surroundings. The total enthalpy of this system is $N_d h^*_{d,init} + N_l h^*_{l,init}$ (J). This will now be considered the initial state of the system.

    After removing the membrane the final state of the system occurs after $N_v$ mols of liquid have evaporated and diffused
throughout the volume of dry air to the point of saturation, where of course $N_v \leq N_l$, to ensure that there is enough liquid to achieve saturation. Note: It is possible to calculate $N_v$, because $N_v = N_{v,sat}$, but for the present purposes this is not necessary. The final state now comprises $N_d$ mols of dry air, $N_l - N_v$ mols of pure liquid water and $N_v$ mols of water vapor. For an ideal gas the final pressure is $p_{v,sat}$ (Pa), but for a non-ideal gas the saturated vapor pressure is $f p_{v,sat}$ (Hyland and Wexler , 1983; Goff , 1949), where $f = f(T_K, p_a)$ is termed the enhancement factor and $1 < f < 1.006$ near STP (Hyland and Wexler
, 1983; Nelson and Sauer , 2004). Consequently, the final pressure of the water vapor will exceed $p_{v,sat}$ by a small amount. On the other hand, the final pressure of the dry air, $p_{d,final}$ (Pa), will be slightly less that $p_d$ because the final gas volume of the system will be slightly greater than the initial volume due to the decrease in the volume of liquid with the evaporative loss of $N_v$ mols of liquid. In the present scenario this difference between the final and initial pressures is small: $\approx 0.001 p_d$. Because both $f$ and this relative pressure difference are so small and they tend to compensate for one another, it is reasonable to ignore
both effects and approximate the final total pressure, $p_a$ (Pa), as simply as $p_a = p_d + p_{v,sat}$; meaning that the present purposes evaporation occurring within an isolated system can be considered as an archetypical constant pressure process. Nonetheless, it is also worth emphasizing that, in fact, evaporation in the present isolated system (as well as within the atmospheric surface layer) is neither a constant volume, nor a constant pressure, process. Rather it is a combination or hybrid of the two processes.

    The total enthalpy of the final state of the system is $(N_d + N_v) h_a + (N_l - N_v) h^*_l$; where $h_a$ (Jmol$^{-1}$) is the specific enthalpy
of resulting moist air. But, because of evaporative cooling the temperature of the final state of the system, $T_K$ (273.16 K $< T_K \leq 373.15$ K), is less than $T_{K,init}$. This change in temperature of the system, $\delta T$ (K), is defined as $\delta T = T_K - T_{K,init} < 0$. The appendix examines this temperature difference in more detail. With this last simplification in mind, the change in total enthalpy of the system, $\Delta H_s$ (J), is

$$\Delta H_s = (N_d + N_v) h_a + (N_l - N_v) h^*_l - (N_d h^*_{d,init} + N_l h^*_{l,init}) \tag{3}$$

where $h_a = \chi_d h^*_d + \chi_v h^*_v + I_B$ (e.g., Hyland and Wexler , 1983) and $\chi_d = N_d/(N_d + N_v) = p_d/p_a$ is the dry air molar fraction (mol mol$^{-1}$) of the moist air, $\chi_v = N_v/(N_d + N_v) = p_{v,sat}/p_a$ is the vapor molar fraction (mol mol$^{-1}$) of the moist air, and $I_B$ is the excess enthalpy of mixing (e.g., Wormald et al. , 1977; Sattar , 2000) that arises because of the non-ideality of the gases (e.g., Hyland and Wexler , 1983).

    After some algebraic manipulation the following simplified expression for $\Delta H_s$ results:

$$\Delta H_s = N_v(h^*_v - h^*_l) + (N_v + N_d) I_B + N_d \delta h^*_d + N_l \delta h^*_l \tag{4}$$

where $\delta h_d^* = h_d^* - h_{d,init}^*$ and $\delta h_l^* = h_l^* - h_{l,init}^*$. Because both $h_d^*$ and $h_{d,init}^*$ are functions of temperature, i.e., $h_d^* = h_d^*(T_K)$, $h_{d,init}^* = h_{d,init}^*(T_{K,init})$ and that $\delta T$ is small in comparison to either $T_{K,init}$ or $T_K$ (Appendix A), it is reasonable to approximate $\delta h_d^*$ as $(\partial h_d^* / \partial T_K)\delta T$. Similar results hold for $\delta h_l^*$. So that the $N_d \delta h_d^* + N_l \delta h_l^*$ component of $\Delta H_s$ can be reasonably assumed to be a function of both temperature and the temperature difference. On the other hand, the $N_v(h_v^* - h_l^*) + (N_v + N_d)I_B$

component of $\Delta H_s$ is a function only of the final temperature, $T_K$, and is not influenced by $\delta T$. This allows the following identification to be made: $\Delta H_s = \Delta H_{s,L} + \Delta H_{s,T}$; where $\Delta H_{s,L} = N_v(h_v^* - h_l^*) + (N_v + N_d)I_B$ results from the change of phase associated with evaporation and $\Delta H_{s,T} = [N_d(\partial h_d^* / \partial T_K) + N_l(\partial h_l^* / \partial T_K)]\delta T$ results from the change in temperature. The principal interest of this study is $\Delta H_{s,L}$. Therefore, dividing $\Delta H_{s,L}$ by $N_v$ yields

$$L_v \equiv \frac{\Delta H_{s,L}}{N_v} = L_v^* + \frac{I_B}{\chi_v} \qquad (5)$$

At this point it is important to note that except for the non-ideality of water vapor and dry air the enthalpy of vaporization of water would be completely independent of the presence of dry air, i.e., $L_v \equiv L_v^*$. In other words, if not for the non-ideal behavior of these gases $L_v$ would be the sole property of water and would otherwise not be influenced by the present or absence of dry air.

In general $I_B$ is expressed in terms of the second and third virial coefficients (Hyland and Wexler , 1983; Wagner and Pruß,

2002), which are defined by the virial equation of state (Hyland and Wexler , 1983; Sattar , 2000) as follows:

$$\frac{p_i v_i}{R T_K} = 1 + \frac{B_i}{v_i} + \frac{C_i}{v_i^2} + \cdots \qquad (6)$$

where the subscript '$i$' refers to water vapor ($i = v$), dry air ($i = d$), or moist air ($i = a$); $B_i$ (m$^3$mol$^{-1}$) is the second virial coefficient, $C_i$ (m$^6$mol$^{-2}$) is the third virial coefficient, and in general $B_i$ and $C_i$ are both functions of temperature, $T_K$; $p_i$ is the gas pressure (Pa) and $v_i$ is the molar volume (m$^3$mol$^{-1}$) of the gas. For this study it is sufficient to consider only the

135 second virial coefficients. For dry air and water vapor $B_i = B_i(T_K)$ is determined by empirical curve fitting of observed data. For this study $B_v(T_K)$ is taken from Equation (6) of Harvey and Lemmon (2004) and $B_d(T_K)$ is taken from Equation (10) of Hyland and Wexler (1983). Because moist air is a mixture of dry air and water vapor the second virial coefficient for moist air takes the form $B_a = \chi_v^2 B_v + 2\chi_v\chi_d B_{vd} + \chi_d^2 B_d$ Sattar (2000), where $B_{vd}$ (m$^3$mol$^{-1}$) is the cross virial coefficient for moist air. For the present study $B_{vd}(T_K)$ is taken from Equation (15) of Hyland and Wexler (1983). Once the equation of state has

140 been specified, the general expression for $I_B$ can be derived (e.g., Sattar , 2000), yielding

$$\frac{I_B}{\chi_v} = p_a\chi_d \left[ 2\left(B_a - T_K\frac{dB_a}{dT}\right) - \left(B_d - T_K\frac{dB_d}{dT}\right) - \left(B_v - T_K\frac{dB_v}{dT}\right) \right] \qquad (7)$$

The final step is to specify whether the enthalpic change occurs at constant pressure or at constant volume. Although assuming a constant pressure pathway for modeling evaporation into the atmosphere is likely to be more appropriate than assuming a constant volume pathway, both pathways need to be considered here because any evaporation occurring on

the earth's surface is going to lie somewhere between these two (bounding) pathways. This is equivalent to specifying $p_a$ and $p_d$ at the initial and final states. At a constant pressure $p_a$ is held constant, so that $p_d$(final state) = $p_d$(initial state) − $p_v$(final state); where $p_d$(initial state) = $p_a$ and $p_v$(final state) = $p_{v,sat}$ and (for the sake of completeness it should also be noted that) $p_v$(initial state) = 0. In this case $p_a$ is arbitrarily assigned a value of 101.325 kPa. To evaluate $L_v^*$ at a constant volume $p_d$ is held constant, so $p_a$(final state) = $p_a$(initial state) + $p_v$(final state). In this case $p_d$ is arbitrarily assigned a value

of 101.325 kPa. The only difference between these two cases is that the final molar values of $N_v$ and $N_a$ (= $N_v + N_d$) can be different, so that the term $p_a\chi_d$ in Equation (7) can vary slightly depending on whether the evaporation is occurring at a constant pressure or a constant volume.

     The results of evaluating Equation (7) for these two different processes are shown in Figure 2. Note that beginning with this figure and henceforth $\Delta L_v$ will be used as shorthand for $I_B/\chi_v$. These results suggests that surface energy fluxes associated

with ET measured at temperatures commonly encountered with micrometeorological techniques (i.e., between about 275 and 315 K) could be underestimated by 1% to 2% solely on the basis of using an estimate for the enthalpy of vaporization, $L_v^*$, that does not allow for the fact that dry air and water vapor are non-ideal gases. Categorically then this underestimate is at least an order of magnitude less that the often observed surface energy imbalance mentioned in the introduction.

## 2.2   Specific Heat

But in many micrometeorological studies of the SEB $L_v E$ is only half the story. There is also the sensible or convective heat flux, $H = \rho_a C_{pa}\overline{w'T'} \equiv \varrho_a c_{pa}\overline{w'T'}$; where $\rho_a$ (kgm$^{-3}$) and $\varrho_a$ (molm$^{-3}$) are the density of the ambient moist air (in mass or molar units) and $C_{pa}$ (Jkg$^{-1}$K$^{-1}$) and $c_{pa}$ (Jmol$^{-1}$K$^{-1}$) are the specific heat of moist air at constant pressure (in units corresponding to the moist air density). $\overline{w'T'}$ is the kinematic heat flux, which is obtained directly from eddy covariance measurements. Assuming ideal gases, $c_{pa}^* = c_{pa}^*(T_K) = \chi_v c_{pv}^*(T_K) + \chi_d c_{pd}^*(T_K)$ is the weighted sum of the specific heats of

pure water vapor (subscript $v$) and pure dry air (subscript $d$). For the present study $c_{pa}^*$, $c_{pv}^*(T_K)$, and $c_{pd}^*(T_K)$ are obtained from Equation (6) of Bücker et al. (2003).

     Using $\Delta$ to denote the departure from ideality, the derivation of $\Delta c_{pa} \equiv \chi_v \Delta c_{pv} + \chi_d \Delta c_{pd}$ begins with the following (standard thermodynamic) relation $dL_v/dT = c_{pv} - c_{pl}$ (e.g., Curry and Webster , 1999, equation 4.29), where $c_{pv}$ and $c_{pl}$ are the specific heats at constant pressure for water vapor (subscript $v$) and liquid water (subscript $l$). Combining this relationship, which is valid for both ideal and non-ideal gases, with Equation (5), it is straightforward to show that

which is valid for both ideal and non-ideal gases, with Equation (5), it is straightforward to show that

$$c_{pv} - c_{pl} = c_{pv}^* - c_{pl}^* + \frac{d(I_B/\chi_v)}{dT} \tag{8}$$

For the present purposes it can be assumed that liquid water always remains pure (or ideal) and therefore, $c_{pl} = c_{pl}^*$. Then identifying $\Delta c_{pv}$ as $c_{pv} - c_{pv}^*$ and using Equation (7) above it follows from Equation (8) that

$$\chi_v \Delta c_{pv} = \frac{dI_B}{dT} = -p_a \chi_d \chi_v T_K \left[ 2\frac{d^2 B_a}{dT^2} - \frac{d^2 B_d}{dT^2} - \frac{d^2 B_v}{dT^2} \right] \tag{9}$$

To complete the estimate of $\Delta c_{pa}$ it is necessary to determine $\chi_d \Delta c_{pd}$, which is easily deduced from the dry air term $(d^2 B_d / dT^2)$ in Equation (9). In this last equation, as well as in Equations (5) and (7), the dry air term (any $B_d$ term) is basically meant to account for the effects of dry air interacting with itself. Consequently, it is fairly straightforward to conclude from Equation (9) that

$$\chi_d \Delta c_{pd} = -p_a \chi_d T_K \left[ \frac{d^2 B_d}{dT^2} \right] \tag{10}$$

Combining this last expression for $\chi_d \Delta c_{pd}$ with that for $\chi_v \Delta c_{pv}$ yields the final results for $\Delta c_{pa} / c_{pa}^*$ as a function of $T_K$, which is shown in Figure 3 overlaying $\Delta L_v / L_v^*$ from Figure 2.

### 2.3   Consequences to the surface energy balance

Implications to the SEB of mixing the two non-ideal gases (water vapor and dry air) during evaporation can now be estimated by combining the results for $\Delta L_v / L_v^*$ and $\Delta C_p / C_p$. For example, assuming a Bowen ratio of approximately unity (i.e., the 185 magnitude of $H$ and $L_v E$ are approximately the same) and a temperature between say 280 K and 350 K, then the term $L_v E + H$ in the SEB could be underestimated between 1% and 1.5% with micrometeorological techniques due to the non-ideality of water vapor and dry air. Allowing for different values of the Bowen ratio would imply a somewhat broader range of percentage underestimates. But even so, it is unlikely that non-ideality could cause $L_v E + H$ to be underestimated by more than 2%, which, at best, is an order of magnitude less than required to account for the imbalance of the SEB.

### 3  $p \Delta v$ and the Surface Fluxes of Sensible Heat and Water Vapor

This section examines the issue Paw U et al. (2000) address, viz., the "energy associated with evaporation into the atmosphere, necessary for the expansion of eddy parcels against an approximately constant pressure". In essence the authors are proposing a correction to eddy covariance measurements of turbulent temperature fluctuations ($T'$) that account for the density change of an air parcel associated with the mixing of a relatively dense fluid (ambient air), with a relatively less dense fluid (water vapor). 195 The following is a slight reformulation of their approach.

    For an adiabatic process the first law of thermodynamics can be expressed as

$$c_v dT + p_a dv_a = 0 \tag{11}$$

where $c_v$ ($\mathrm{Jmol^{-1}K^{-1}}$) is the molar specific heat of moist air at constant volume and $v_a$ ($\mathrm{m^3 mol^{-1}}$) is the specific volume of air, which by definition is the reciprocal of the molar air density, $\varrho_a$ ($\mathrm{molm^{-3}}$). Switching from differential notation to

perturbation notation, Equation (11) can be written as $c_v T' + p_a v'_a = 0$. By definition $v_a = 1/\varrho_a$ so it also follows that $v'_a = -\varrho'_a/\varrho_a^2$, which combined with the ideal gas law $p_a v_a = RT_K$ yields the following equivalent expression for Equation (11):

$$T'_e + \frac{RT_K}{c_v}\frac{v'_a}{v_a} = 0 \quad \text{or} \quad T'_e = \frac{RT_K}{c_v}\frac{\varrho'_a}{\varrho_a} \tag{12}$$

where $T'_e$ is defined by Paw U et al. (2000) as "the temperature perturbation equivalent to the energy needed for expansion". Next they assume that the change in molar air density, $\varrho'_a$, is due to the mol per mol displacement of moist air by water vapor,

so that for present purposes $\varrho'_a = -\varrho'_v$, from which is follows that

$$T'_e = -\frac{RT_K}{c_v}\frac{\varrho'_v}{\varrho_a} \equiv -\frac{\mu RT_K}{c_v}\frac{\rho'_v}{\rho_a} \tag{13}$$

where the second expression on the right is expressed in mass units (kg) rather than mols, i.e., $\mu = 1.609$ is the ratio of the molecular mass of dry air to the molecular mass of water vapor, $\rho_v$ ($\text{kgm}^{-3}$) is the mass density of water vapor, and $\rho_a$ ($\text{kgm}^{-3}$) is the mass density of the ambient atmosphere. Equation (13) is a rephrasing of the principal result – Equation (C3)

– of Appendix C of Paw U et al. (2000).

     Before proceeding with an alternative approach to deriving an expression for $T'_e$, it is insightful to examine an apparent sign error made by Paw U et al. (2000) in their mathematical development of $T'_e$ and its equivalent heat flux – their Equation (C4). First, they pose their Equation (C2) as the antecedent to their expression for $T'_e$, by asserting that $v'_a \propto -\rho'_v$ when displacing heavier dry air molecules by lighter water vapor molecules; meaning that the specific volume perturbation should decrease, i.e.,

$v'_a < 0$. But this contradicts the fact that the specific volume should increase when the density of the (formerly dry) air parcel decreases when displacing heavier molecules by lighter ones. From the discussion in the paragraph immediately preceding the present one – $v_a = 1/\varrho_a$ implies $v'_a = -\varrho'_a/\varrho_a^2$ combined with the displacement assumption $\varrho'_a = -\varrho'_v$ – it follows that $v'_a \propto \rho'_v > 0$, in agreement with expectations. Interestingly, despite this sign error in Equation (C2), Paw U et al. (2000) have the same sign for their $T'_e$ – their Equation (C3) – as Equation (13) above, i.e., both expressions yield $T'_e < 0$. Nonetheless and

even more puzzling is that Paw U et al. (2000) reverse the sign again when they proceed to their Equation (C4), the succedent to their Equation (C3). In this step of their development the heat flux $H_e$, generally defined such that $H_e \propto T'_e < 0$, they suggest $H_e \propto -T'_e > 0$. The reason for reversing the sign a second time is not discussed, nor is how this might relate to the pressure work term. But if their goal is to determine an equivalent heat flux associated with a change in density of an air parcel due to the partial displacement of a heavier gas in that air parcel by a lighter gas, then it is reasonable to expect that $H_e > 0$ because that air parcel would be positively buoyant relative to the surrounding (drier and heavier) air. Assuming this conjecture is true,

then this contradiction in the sign of $H_e$ suggests seeking an alternative approach to determine $H_e$. The remaining portion of this study outlines such an alternative.

     The final portion of this study attempts to clarify the nature of $H_e$ and the role of the work term and whether the surface sensible heat flux includes a water vapor term similar to that suggested by Paw U et al. (2000) and recast as Equation (13)

above. I begin with the time dependent version of the first law of thermodynamics expressed as the conservation law for potential temperature, $\theta$ (K), for an incompressible atmospheric process:

$$\frac{d\theta}{dt} = \frac{\partial\theta}{\partial t} + \nabla(\mathbf{u}\theta) = \frac{1}{c_p}\frac{\theta}{T_K}\frac{dq}{dt} \tag{14}$$

where $dq/dt$ represents the heat flow associated with diabatic atmospheric processes, $\mathbf{u}$ (ms$^{-1}$) is the atmospheric velocity, $\nabla$ is the vector gradient operator, and the vector dot product has been dropped for convenience. Equation (14) is standard and

235 in and of itself is not novel. Nonetheless it does imply an important take away: which is that the turbulent surface sensible heat flux is more correctly expressed in terms of potential temperature, $\overline{w'\theta'}$, rather that in terms of temperature, $\overline{w'T'}$. This is principally because $\overline{w'\theta'}$ explicitly includes the effects of any change in ambient pressure and the concomitant work done on or to the atmosphere during turbulent atmospheric processes. NOTE: for the sake of completeness $w$ (ms$^{-1}$) is the vertical velocity and the $'$ notation is standard and refers to Reynolds averaging. Having identified potential temperature as the key

variable for discussion the next step is to examine the influence moisture has on $\theta$.

Including the effects of water vapor on potential temperature yields the following relation (e.g., Curry and Webster , 1999).

$$\theta = T_K \left(p_{00}/p_a\right)^{\kappa(1-0.33q_v)} \equiv T_K \left(p_{00}/p_a\right)^{\kappa} e^{-0.33\kappa q_v \log(p_{00}/p_a)} \tag{15}$$

where $p_{00} = 100$ kPa is a constant reference pressure; $\kappa = R_d/C_{pd}$, for which $C_{pd}$ is the specific heat for dry air and consequently $\kappa = 2/7$ is an extremely good approximation; and $q_v = \rho_v/\rho_a$ (kgkg$^{-1}$) is the specific humidity of moist air. Note: that

the 0.33 coefficient modifying $q_v$ corrects what appears to this author to be an error in Equation (2.66) and related expressions in Curry and Webster (1999). Equation (15) clearly indicates that $\theta$ is dependent on moisture. Although this dependency is extremely weak, the purpose here is to assess the influence of $\rho'_v$ on the $\theta'$ using Equation (15) and to compare the result with Equation (13). A sketch of the derivation follows.

Linearize Equation (15) first by noting that near-surface atmospheric conditions (i.e., $q_v < 0.04$ and $\log(p_{00}/p_a) < 0.35$ or

250 $0.33q_v < 0.014$ and $\kappa\log(p_{00}/p_a) < 0.1$) are sufficient to guarantee that $0.33\kappa q_v \log(p_{00}/p_a) < 0.33q_v \ll 1$ and second by assuming that the perturbation quantities are small compared to their background levels (which will be denoted by an overbar). This yields:

$$\frac{\theta'}{\overline{\theta}} = \frac{T'}{T_K} - \kappa\frac{p'_a}{p_a} - \alpha' \tag{16}$$

where $\alpha = 0.33\kappa q_v \log(p_{00}/p_a)$, $\alpha' = -0.33\kappa\overline{q_v}(p'_a/\overline{p_a}) + 0.33\kappa\log(p_{00}/\overline{p_a})q'_v$, and for later use $\overline{\alpha} = 0.33\kappa\overline{q_v}\log(p_{00}/\overline{p_a})$.

Substituting $\alpha'$ into Equation (16) yields

$$\frac{\theta'}{\overline{\theta}} = \frac{T'}{T_K} - (1-0.33\overline{q_v})\kappa\frac{p'_a}{p_a} - \gamma q'_v \tag{17}$$

where $\gamma = 0.33\kappa \log(p_{00}/\overline{p_a}) < 0.034 \ll 1$. Next is the evaluation of $q'_v$ by expanding and linearizing $q_v = \rho_v/\rho_a$ in terms of $\rho'_v$ and $\rho'_a$. This yields $q'_v = \rho'_v/\overline{\rho_a} - \overline{q_v}(\rho'_a/\overline{\rho_a})$. The ideal gas law for ambient air yields $\rho'_a/\overline{\rho_a} = p'_a/\overline{p_a} - T'/\overline{T_K}$ and therefore, $q'_v = \rho'_v/\overline{\rho_a} - \overline{q_v}(p'_a/\overline{p_a}) + \overline{q_v}(T'/\overline{T_K})$. Substituting this last expression for $q'_v$ into Equation (17) yields:

$$\theta' = \overline{\theta}(1-\overline{\alpha})\frac{T'}{\overline{T_K}} - \overline{\theta}(1-\beta)\kappa\frac{p'_a}{\overline{p_a}} - \overline{\theta}\left(\gamma\frac{\rho'_v}{\overline{\rho_a}}\right) \tag{18}$$

or

$$\theta' = \overline{\theta}\left(\frac{T'}{\overline{T_K}} - \kappa\frac{p'_a}{\overline{p_a}}\right) - \overline{\theta}\left(\overline{\alpha}\frac{T'}{\overline{T_K}} - \beta\kappa\frac{p'_a}{\overline{p_a}}\right) - \overline{\theta}\left(\gamma\frac{\rho'_v}{\overline{\rho_a}}\right) \tag{19}$$

where $\beta = 0.33\overline{q_v}[1 + \log(p_{00}/\overline{p_a})] < 0.018 \ll 1$. At this point it is important to reiterate that for near-surface conditions $\alpha < 0.33\overline{q_v} < \beta < 0.018 \ll 1$.

Equation (19) suggests that water vapor contributes two different "corrections" to the kinematic heat flux. First, the middle term on the right hand side of this equation, is due to the overall presence of water vapor, $\overline{q_v}$, and the second, the last term on the right hand side of Equation (19) and the term of interest in this study, results from fluctuations in water vapor, $\rho'_v$. Although Equations (13) and (19) have somewhat different definitions of heat flux, it is still possible to assess the appropriateness of the displacement assumption made by Paw U et al. (2000) by numerically comparing the dimensionless coefficient $\mu R/c_v$ in Equation (13) with $\gamma$ in Equation (19). Noting that $R/c_v = 2/5$, then $\mu R/c_v \approx 0.644 \approx 19\gamma$. In other words $\gamma \ll \mu R/c_v$ and therefore, the approach followed by Paw U et al. (2000) predicts significantly more turbulent heat flux associated with the water vapor flux than does the approach based on potential temperature (initiated above with Equation (14)). Even allowing for the difference between potential temperature and $T_K$ does not really change this result by more than 10% because $(p_{00}/p_a)^\kappa < 1.1$ for conditions being considered here.

This difference between Paw U et al. (2000) and the present result is made more explicit by comparing the next two expressions. The first expression derives from combining Equation (13) for $T'_e$ with the equivalent heat flux, $H_e = \overline{\rho_a}C_p\overline{w'T'_e}$ from Paw U et al. (2000). This yields the following generalization of Paw U et al. (2000) result:

$$H_e = -\frac{\mu C_p}{c_v}\left(\frac{RT_K}{L_v}\right)L_vE \approx -\frac{9}{4}\left(\frac{R_dT_K}{L_v}\right)L_vE \approx -(0.07 - 0.10)L_vE \tag{20}$$

where $R_d = 287 \text{ Jkg}^{-1}\text{K}^{-1}$. The second expression results by identifying the equivalent potential temperature, $\theta'_e$, associated with the water vapor perturbation, $\rho'_v$, in Equation (19), i.e., $\theta'_e = -\overline{\theta}(\gamma\rho'_v/\overline{\rho_a})$ and combining it with the expression for the equivalent heat flux, $H_e$, appropriate to Equation (14), i.e., $H_e = \overline{\rho_a}C_p\overline{w'\theta'_e}$. This yields

$$H_e = -\left(\frac{\gamma C_p\overline{\theta}}{L_v}\right)L_vE \approx -\frac{3}{25}\left(\frac{R_dT_K}{L_v}\right)L_vE \approx -(0.0037 - 0.0053)L_vE \tag{21}$$

It is not possible to reconcile these two expressions, which brings into question the validity of the displacement assumption of Paw U et al. (2000) (i.e., $\varrho'_a = -\varrho'_v$), on which Equation (20) is based. But, to truly assess the cogency of this assumption

and any enthalpic changes associated with mixing of the dry air and water vapor requires a better description of the physical processes and the initial and final states involved than Paw U et al. (2000) provide. Nevertheless, since they are addressing evapotranspiration, it seems reasonable to assume they are envisioning the final state of the evaporative process. In this case the work done to/by the atmosphere associated with the expansion of water vapor into the atmosphere is appropriately included in the enthalpy of vaporization as previously discussed and, consequently, the displacement assumption would result in over-counting the work term. On the other hand, if they are describing the enthalpic changes associated with rising plumes of warm moist air associated with density differences between very moist air near the surface and drier and therefore, denser air above the near-surface, then the methods and results outlined by Equations (14), (15), and (21) above are more appropriate.

## 4   Conclusions

The present study has explored some of the issues involving the surface energy balance (SEB) and the thermodynamics of evaporation of water into the atmosphere. Specifically I have looked at (a) the influence that molecular interactions between water vapor and dry air (non-ideality of atmospheric gases) could have on estimates of $L_v$ and $C_p$ and the SEB and (b) the impact that fluctuations of atmospheric water vapor could have on the surface heat flux. At typical atmospheric temperatures (285-325 K), the influence of the first affect is probably on the order of about 1% and the second is about 0.5%. Consequently, these phenomena acting either independently or in consort are far too small to be of any real significance in explaining the lack of closure of the SEB. This result should not be surprising, but because these issues may not be well known to the micrometeorological and geo-biophysical communities it seemed worthwhile to attempt to verify this supposition quantitatively.

*Code and data availability.*   The computer code used in this study was developed using MatLab version 2017b and is publicly available along with any output data at the Forest Service Research Data Archive https://doi.org/10.2737/RDS-2019-0042. Prior to its availability online the code and any output is freely available from the author.

## Appendix A

This appendix derives the relationship between $\delta T$ and $N_v$, $N_d$ and $N_l$ appropriate to evaporative cooling of the isolated thermodynamic system discussed in section 2.1 of the main text. Achieving this requires an approach similar to that used when calculating the wet bulb temperature (e.g., Curry and Webster , 1999). The formal expression for the first law of thermodynamics for the system under consideration is:

$$0 = Q_f - Q_i = (C_{ps}T_K)_f - (C_{ps}T_K)_i + (N_v M_v L_v^*)_f - (N_v M_v L_v^*)_i \tag{A1}$$

where the subscripts $f$ and $i$ refer to the final and initial states; $Q_f - Q_i$ (J) is the total heat exchanged by the system and its environment, which must be 0 since the system is isolated from its environment; $C_{ps}$ (JK$^{-1}$) is the bulk heat capacity of

the composite system (vapor + dry air + and pure liquid water) at constant pressure so that the change in heat content of the system, $(C_{ps}T_K)_f - (C_{ps}T_K)_i$, must exactly cancel the change in the enthalpy of the system $(N_v M_v L_v^*)_f - (N_v M_v L_v^*)_i$, which is expressed here in terms of the water vapor component. $L_v^*$ assumes that water vapor is ideal gas (an assumption that is sufficient for the present purposes) and $M_v$ (kgmol$^{-1}$) the molecular mass of water vapor. Simplifying this expression begins by identifying

$$T_{Kf} = T_{Ki} + \delta T \tag{A2}$$

and

$$C_{ps} = N_d M_d c_{pd} + N_v M_v c_{pv} + (N_l - N_v) M_l c_{pl} \equiv N_d M_d c_{pd} + N_l M_v c_{pl} + N_v M_v (c_{pv} - c_{pl}) \tag{A3}$$

where $N$ refers to the number of mols of any particular component (subscript $d$ for dry air; $v$ for vapor, and $l$ for liquid); $M$ refers to the molecular mass of that component; $c_p$ (Jkg$^{-1}$K$^{-1}$) refers to the specific heat at constant pressure of that component; and $M_l = M_v$ has been used for the right hand side of the last expression.

Combining these last two expressions with $N_{vi} = 0$ and after dividing the resulting expression by $M_v$ yields;

$$\left( \frac{N_d M_d}{M_v} c_{pd} + N_l c_{pl} + N_v (c_{pv} - c_{pl}) \right) \delta T = -N_v (c_{pv} - c_{pl}) T_{Ki} - N_v L_v^* \tag{A4}$$

where the $\delta T$ term on the left hand side and the term $N_v L_v^*$ on the right hand side are evaluated at $T_K$ (the final temperature) and the first term on the right hand side, $-N_v(c_{pv} - c_{pl})T_{Ki}$, is evaluated at the initial temperature $T_{Ki}$. The order of magnitude calculation is facilitated by dividing the last expression by $c_{pl}T_{Ki}$, by noting that $M_d/M_v \approx 1.6$ and that $c_{pl} \approx 2c_{pv} \approx 4c_{pd}$, and by ignoring the relatively weak temperature dependency of the various $c_p$'s. This yields:

$$(0.4N_d + N_l - 0.5N_v) \frac{\delta T}{T_{Ki}} = \left( 0.5 - \frac{L_v^*}{c_{pl}T_{Ki}} \right) N_v \tag{A5}$$

The last step to deriving $\delta T = \delta T(N_v, N_d, N_l)$ assumes that $L_v^* \approx 2.5 \times 10^6$ Jkg$^{-1}$ and requires noting that for the temperature range 295 K $\leq T_{Ki} \leq$ 325 K, $c_{pl} \approx 4.186 \times 10^3$ Jkg$^{-1}$K$^{-1}$ and $c_{pl}T_K \approx (1.24-1.36) \times 10^6$ Jkg$^{-1}$. These last conditions yield the final result:

$$\frac{\delta T}{T_{Ki}} \approx -1.4 \left( \frac{N_v}{0.4N_d + N_l - 0.5N_v} \right) \tag{A6}$$

For the isolated thermodynamic system discussed in the present study it is also valid to assume that $N_v \ll N_d$, which in turn is sufficient to guarantee that $\delta T \ll T_{Ki}$, meaning that the temperature change associated with evaporative cooling should be quite small.

*Author contributions.* W. J. Massman is the sole author of this paper.

*Competing interests.* The author declares no competing interests.

*Acknowledgements.* The author would like to thank Allan Harvey for his insights and his significant time and efforts involving section 2 of this work. I also gratefully acknowledge Kyaw Tha Paw U, Grant Petty, Jim Wilczak, and Ned Patton for their comments during the development of this work.

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

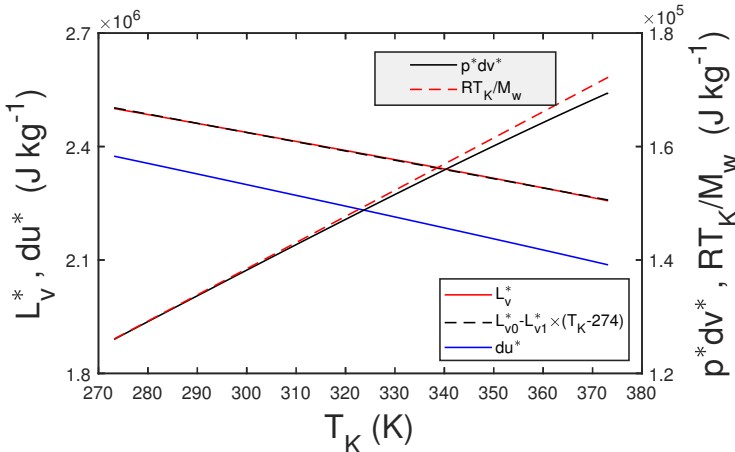

**Figure 1.** The solid red line is the enthalpy of vaporization of pure water, $L_v^*$, from Wagner and Pruß (2002), and the dashed black line is a linear approximation to it; where $L_{v0}^* = 3.16924 \times 10^6$ Jkg$^{-1}$ and $L_{v1}^* = 2.4405 \times 10^3$ Jkg$^{-1}$K$^{-1}$. The solid blue line is the thermodynamic change in internal energy, $du^*$, associated with $L_v^*$. The other two lines are $p^* \Delta v^*$ (solid black) and $RT_K/M_w$ (dashed red) after multiplication by a factor of 15 for ease in plotting and visualization. Here $T_K$ is the temperature in degrees K, $R$ (Jmol$^{-1}$K$^{-1}$) is the universal gas constant, and $M_w = 0.0180153$ kgmol$^{-1}$ is the molecular mass of water vapor.

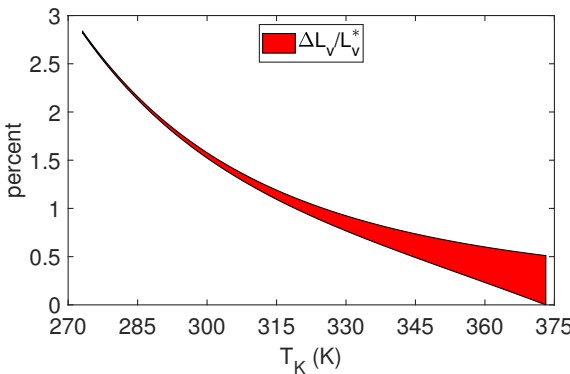

**Figure 2.** % change in $L_v^*$ from Equation (7), where $\Delta L_v$ on the y-axis label is used in place of $I_B/\chi_v$. The upper curve bounding the red shaded area is $\Delta L_v/L_v^*$ for a process occurring at constant volume and the lower boundary is for a isobaric process.

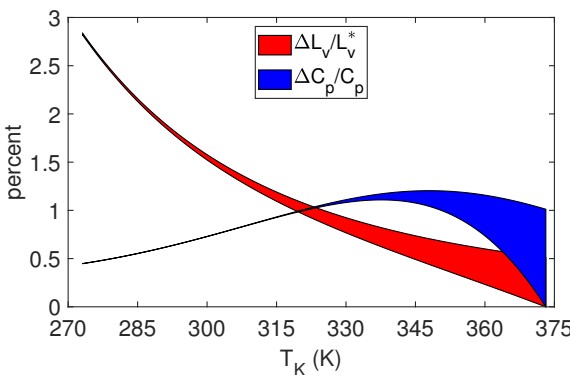

**Figure 3.** % change in $C_p$ from the sum of Equations (9) and (10) along with (overlaying) the results for $\Delta L_v/L_v^*$ as shown in Figure 2. The upper curve bounding the blue shaded area is $\Delta C_p/C_p$ for a process occurring at constant volume and the lower boundary is for a isobaric process.