# Peer review of "Impacts of non-ideality and the thermodynamic pressure work term $p\Delta v$ on the Surface Energy Balance"

_Hydrology and Earth System Sciences, 2019_

## Referee Comment (RC1) · Anonymous Referee #1 · 18 Jun 2019

**Review of HESS Manuscript #2019-153**

Title:   Impacts of non-ideality and the thermodynamic pressure work term pΔv on the Surface Energy Balance

Author: Massman

**Review**

This paper describes how latent heat of vaporisation and a missing pdV terms might contribute to energy imbalance in global flux databases. It is an old fashioned approach in that the conclusion is that neither of the considered corrections can make any major difference. This is a return to the days when negative results were routinely published. It is very useful!

I had some technical comments on the manuscript but nothing major and all should be easily fixed by the author.

**Other Comments**

1.      Formatting of references. On the first few pages, every time the U. Paw or Kowalski studies were cited, there were a number of extra symbols that clearly did not belong. Some sort of font conversion problem I assume. For example, p. 2, line 2 was the first but there are many instances after that. I leave it to you to sort out.

2.      p. 2, lines 15-20. The mechanism/s discussed here are almost identical to those discussed by Makarieva et al (2013). Even more interesting was the controversy generated by the Makarieva et al (2013) paper. See for example: https://judithcurry.com/2013/01/31/condensation-driven-winds-an-update-new-version/ written from the point of view of the authors. More generally, it might be useful to frame the mechanism in terms of Makarieva et al (2013).

3.      p. 3, lines 9-10. "…..… specific enthalpy of pure water". In this context by water you mean liquid water. Given the overall context and the precision of this work, it might be better to call this …. pure **liquid** water ……. Perhaps adopt that convention throughout.

4.      p. 3, line 17 & Figure 1. In the text it states, "….. overlays the blue line ….". But in Figure 1, the red line overlays a dashed black line so this needs to be fixed. More generally, I found Fig. 1 very hard to understand. Instead why not have a secondary axis on the right hand side, or another panel.

5.      p. 3, lines 15-20 & Figure 1. I did not see the advantage of including the change in internal energy du since it is not explained ? The origin of the $p^*\Delta v^*$ (or $RT_k/M_w$ ) term is also not explained in sufficient detail. Here I suggest expanding this section to include equations for internal energy (and enthalpy) and to show the origin of the $p^*\Delta v^*$ term (and the $RT_k/M_w$ term as well). Without this explanation this is very hard to follow for a reader not experienced with classical thermodynamics.

6.      P. 3, line 21. Range should be from the triple point, i.e., 273.16 K (not 273.15 K).

7.      P. 3, line 20- p. 4, line 4. Why start with a liquid-dry air system and then add vapour? This is hard to understand and is non-physical sdince it is hard to imagine starting with liquid in contact with dry air but the liquid is not evaporating. Instead you could start with a liquid-vapor equilibrium and then add dry air which is easy to do experimentally and MUCH easier for a reader to visualise. For example, in your terminology, the initial state is $N_v h_v^* + N_l h_l^*$ and you add dry air to increase the gas pressure from the initial vapour pressure to the standard pressure (1 bar, or I note you use 1 atm). The final state then has an additional term for the dry air and depending on how the change has been carried out (e.g., constant T, P, V, etc..) you will not have to worry about the evaporative cooling. Have I missed something here?

8.      Eqn 2. I can see you have used a different symbol, curly H, for the change. But why not $\Delta h$ instead? That would be MUCH easier for a reader to understand.

9.      P. 5, line 4. I did not understand the statement about ……. enthalpic change at constant pressure or constant volume? Have I missed something here? The change in enthalpy is TdS + VdP. At constant pressure, the enthalpic change is just the heat flux (= TdS) and enthalpy is often called the heat function for constant pressure. By similar logic, the internal energy is the heat function for constant volume? I think what is meant here is whether the process is carried out at constant pressure or constant volume. Correct?

I hope these suggestions prove useful.

**References**

Makarieva A.M., Gorshkov V.G., Sheil D., Nobre A.D., Li B.-L. (2013) Where do winds come from? A new theory on how water vapor condensation influences atmospheric pressure and dynamics. Atmospheric Chemistry and Physics, 13, 1039-1056.

---

## Referee Comment (RC2) · Andrew Kowalski (Referee) · 19 Jun 2019

Review of HESS Manuscript #2019-153

Title: Impacts of non-ideality and the thermodynamic pressure work term pΔv on the Surface Energy Balance

Author:Massman

Review by A. S. Kowalski

This manuscript examines two thermodynamic issues that the author intends to improve the surface energy balance at the margins. I applaud Dr. Massman's dedication to this unresolved and important dilemma in micrometeorology. The two issues addressed are, first the relevance of the virial (versus ideal) gas law, and second the definition of the pressure-work term pΔv as regards the role of the water vapour flux on sensible heat exchange. The former, I believe, represents a substantial contribution to the state of knowledge, if a slight adjustment to thermodynamic accounting, and certainly deserves publication. The latter is framed in a way that is based on two previous publications, both of which I believe to be erroneous, and so should be deleted. Consequently, my recommendation is for major revision in order to publish the most valuable aspect of this manuscript, namely the impact of non-ideality on the Surface Energy Balance.

Specific comments

**Page 2, line 2:** I see no benefit to framing the introduction based on these two references, and suggest a restructuring of the introduction along different lines.

The Kowalski citation is to a manuscript that the referees discredited and the author withdrew. It is incorrect, and so *Hydrology and Earth System Sciences* did not publish it. Furthermore Dr. Massman seems to have realised its irrelevance, since his last reference to this manuscript (at page 2, line 23) indicates an intention to present further analysis and discussion, which later did not follow. I recommend not citing such grey literature that failed to pass the peer-review process.

The Paw U paper is cited only regarding its appendix C, which I believe is patently incorrect (see below). I also recommend not citing this paper, for reasons provided below.

**Page 2, line 23:** The author makes inconsistent use of the first person plural/singular (I/we) at different places in the manuscript (compare with line 26 of the same page). It would be best to homogenize this, perhaps taking into account that this is a single author manuscript.

**Page 4, line 4:** The assumption of no change in temperature for the liquid has not been adequately justified. Given that there are million times more moles of liquid ($N_l$) than vapour ($N_v$), the appropriateness of neglecting (in equation 2) a term with the form $N_l (h_{l,final} - h_{l,inital})$ is hardly obvious. Please make more explicit the justification of this assumption.

**Page 4, lines 12-16:** The note put forth in this paragraph seems to be an unnecessary digression, whose elimination would improve the flow of the manuscript.

**Page 5, line 26:** Perhaps you could support the assertion that $C_p = dL_v / dT$ is a definition, either with a citation or an explanation.

**Page 6, line 23:** Equation 8 comes from equation C2 of the Paw U paper, which I believe is in error. Those authors put forth relationship to describe "the density of air change solely from the perturbation in water vapour", yielding a negative proportionality between perturbations in the specific volume and those in the water vapour density ($\alpha' \propto -\rho_v'$). Note that the context of this relationship is the Webb et al. (1980) paper, which assumes constant pressure. Excluding temperature effects (the *other* WPL correction), the effect of isobaric and isothermal evaporation is to humidify the air. According to the ideal gas law (equation 4 of Webb et al. (1980)), under such conditions the total number of molecules per unit volume remains constant, such that – for a fixed Eulerian volume – dry air molecules disappear at the same rate that water vapour molecules appear due to humidification. Since water vapour has less mass than does dry air, the effect of such humidification is a reduction in air density, and hence an increase in the specific volume, demonstrating that the proportionality between perturbations in the specific volume and those in the water vapour density is in fact *positive*. Therefore, I believe that equation C2 of the Paw U paper is patently incorrect. Since the entirety of Section 3 based on this, I recommend its elimination.

**Page 6, line 24:** This part of the paper (if not eliminated) would be more clear if $T_e'$ were defined more explicitly, as "the temperature perturbation equivalent to the energy needed for expansion" as in the Paw U paper.

---

## Author Comment (AC1) · 24 Jun 2019

William Massman

wmassman@fs.fed.us

**Response to Comments dated 18 June 2019 by Anonymous Referee 1**

(0) My thanks to the referee for his/her comments. My numbered responses correspond to the referee's numbers.

(1) You are correct. I overlooked several formatting errors. I believe I have resolved them all.

[Figure]

(2) The referee makes a very intriguing suggestion. But I cannot fully respond to it without some further study of Makarieva et al.'s hypothesis. I do see some interesting similarities, but they proceed one step further than I by coupling their "model" to an equation of motion (condensation-driven winds). This is a step beyond the intention of this paper. But thanks for drawing my attention to their work. I read the original paper, but have not had occasion to revisit it.

(3) I tried to use phrase "pure liquid water" where I thought the additional "pure" was appropriate.

(4) I have removed the sentence from the text. It was a vestige from the previous version.

(5) I will revise the manuscript to include a discussion of the internal energy and the work terms. I will also revise Figure 1 with a second axis on the right hand side with the correct scaling for $p^*\Delta v^*$.

(6) The temperature was changed to 273.16 K.

(7) (a) The referee is concerned that the initial state of the system is "non-physical". To a certain extent this is true, but this part of the manuscript is really a thought experiment and an exercise in logic designed to elucidate the relationship between non-ideal gases and the enthalpy of vaporization. In fact Equation (4) is the purpose of this exercise. Nonetheless it is possible to make the thought experiment more physically realistic by imagining an impermeable barrier separating the dry air from the liquid, which upon removal would allow evaporation to proceed along its normal course. (Of course there is no such thing as a impermeable barrier because gas

molecules would still manage to diffuse through it.) But in the final analysis this approach is identical to the one that is described in the manuscript.

(b) The referee suggests adding dry-air to a saturated liquid. This approach runs afoul of the assumption that the system is isolated (no exchange of mass or energy between the system and its surroundings). It is not possible to add dry air to an isolated system of liquid water and saturated water vapor. Therefore, the dry air must be part of the system from the beginning, in which case the total enthalpy of the initial state of the system is not $N_v h_v^* + N_l h_l^*$ because it does not include the total enthalpy of the dry air. In a laboratory setting it should be possible to add dry air to a (non-isolated) saturated liquid-vapor system. But then there would be issues of the work term associated with the expansion of dry air into the volume of saturated vapor. In turn this would raise issues concerning the diffusion of the saturated vapor into whatever volume contains the dry air. And then if the vapor pressure drops below saturation because water vapor (now no longer isolated) diffuses out of its original volume, evaporation must begin again to restore the saturated vapor pressure. Implying I now have to contend with evaporative cooling after all. I am not sure that this approach represents any improvement over what is already presented in the manuscript.

(8) I changed the symbol to $\Delta H_s$.

(9) The referee is correct. I am referring to a process occurring at constant volume or constant pressure. Upon rereading the original sentence I see I used "The final step is to specify whether the enthalpic change occurs at **a** constant pressure or at **a** constant volume." The revision removes the two articles, "a". The sentence now reads "The final step is to specify whether the enthalpic change occurs at constant pressure or at constant volume." Sorry for the confusion. Does this address the referee's concern?

---

## Author Comment (AC2) · 9 Aug 2019

**Response to Comments from Andrew Kowalski dated 19 June 2019**

My thanks to Andrew Kowalski for his comments. They were helpful. My response (in italics) follow a repeat of his comment.

**ASK (a)** This manuscript examines two thermodynamic issues that the author intends to improve the surface energy balance at the margins. I applaud Dr. Massman's dedication to this unresolved and important dilemma in micrometeorology. The two

issues addressed are, first the relevance of the virial (versus ideal) gas law, and second the definition of the pressure-work term $p\Delta v$ as regards the role of the water vapour flux on sensible heat exchange. The former, I believe, represents a substantial contribution to the state of knowledge, if a slight adjustment to thermodynamic accounting, and certainly deserves publication.

**RESPONSE (a)** *Thank you for your positive response to the first portion of the paper.*

**ASK (b)** The latter is framed in a way that is based on two previous publications, both of which I believe to be erroneous, and so should be deleted. Consequently, my recommendation is for major revision in order to publish the most valuable aspect of this manuscript, namely the impact of non-ideality on the Surface Energy Balance.

**RESPONSE (b)** *As far as the second half of the paper is concerned. I do not agree that it should be deleted solely on basis of whether the papers I cited were flawed or erroneous. My result in this section can stand on its own merits without reference to Paw U et al. (2000) or Kowalski (2018), which would certainly argue for keeping it regardless of whether other papers have made mistakes or not. I admit that it is possible to reframe the paper, but the paper would lose context.*

**RESPONSE (c)** *These two papers motivated my thinking and interest in this subject and in writing this paper and not to cite them is to avoid giving credit where credit is due. I personally am less interested in errors made than I am in trying to understand the issues and the nature of the physical processes processes they are invoking. In this regard, I have been discussing Appendix C of Paw U et al.'s paper with Paw U off-and-on for much of the last decade and more intensely in the past couple years. I thought Appendix C raised an interesting question that I wanted better to understand. But what provided the final impetus to research this subject was Kowalski (2018) and the comments it generated from Paw U, Petty, and Meester. So I felt that a paper that delved a little deeper into the role of thermodynamics might be of benefit to the community. And I certainly benefited by the effort of researching and writing it.*

[Figure]

Specific comments

**ASK - Page 2, line 2:** I see no benefit to framing the introduction based on these two references, and suggest a restructuring of the introduction along different lines. The Kowalski citation is to a manuscript that the referees discredited and the author withdrew. It is incorrect, and so Hydrology and Earth System Sciences did not publish it. Furthermore Dr. Massman seems to have realised its irrelevance, since his last reference to this manuscript (at page 2, line 23) indicates an intention to present further analysis and discussion, which later did not follow. I recommend not citing such grey literature that failed to pass the peer-review process.

**RESPONSE** *I cited HESS-Discussions for Kowalski (2018), not HESS. I also note that the paper does have a doi and is archived on and retrievable from the journal website. As I explained in my **Response (c)** above I do not agree that the paper is without merit. But I did reword the sentence on page 2, lines 24-25 that the author identifies as being on page 2, line 23.*

**ASK -** The Paw U paper is cited only regarding its appendix C, which I believe is patently incorrect (see below). I also recommend not citing this paper, for reasons provided below.

**RESPONSE** *I agree that there is an obvious contradiction in Paw U et al., but I disagree that it obviates the reason for citing it. I provide further discussions on the point below as well.*

**ASK - Page 2, line 23:** The author makes inconsistent use of the first person plural/singular (I/we) at different places in the manuscript (compare with line 26 of the

same page). It would be best to homogenize this, perhaps taking into account that this is a single author manuscript.

**RESPONSE** *I agree. The text has been changed to "I" throughout (if the "we" wasn't just eliminated instead).*

**ASK - Page 4, line 4:** The assumption of no change in temperature for the liquid has not been adequately justified. Given that there are million times more moles of liquid $(N_l)$ than vapour $(N_v)$, the appropriateness of neglecting (in equation 2) a term with the form $N_l$ $(h_{l,final} - h_{l,inital})$ is hardly obvious. Please make more explicit the justification of this assumption.

**RESPONSE** *The reviewer is correct. I did not properly account for the change in temperature of the system associated with evaporative cooling. I have corrected this error and revised the main text and the appendix accordingly. The revised text now points out that the change in the enthalpy of the system has two components. One associated with the change of phase during evaporation and one associated with the resulting temperature change. For the purposes of this study it is sufficient to focus solely on the first term – the enthalpy of vaporization – in order to estimate the effects of non-ideality of dry air and water vapor on the surface energy balance.*

**ASK - Page 4, lines 12-16:** The note put forth in this paragraph seems to be an unnecessary digression, whose elimination would improve the flow of the manuscript.

**RESPONSE** *I agree the eliminating this paragraph would improve the flow of the paper. But there are many papers that are either devoted to this enhancement factor or actually use it for algorithm development. Maybe it is obvious to others that the enhancement factor, $f$, results from non-ideality of gases, but it was not to me until I started looking into this subject more closely. I would prefer to keep this paragraph.*

**ASK - Page 5, line 26:** Perhaps you could support the assertion that $C_p = dL_v/dT$ is a definition, either with a citation or an explanation.

**RESPONSE** *My assertion that $C_p = dL_v/dT$ (by definition) is in error. In fact, after a few seconds of thought it is obvious that it is a mistake. The revisions correct this misstatement and provide a derivation of the corrected $\Delta C_p$ term. Equation (6) and Figure 3 have also been revised accordingly. But numerically the error is very slight.*

**ASK - Page 6, line 23:** Equation 8 comes from equation C2 of the Paw U paper, which I believe is in error. Those authors put forth relationship to describe "the density of air change solely from the perturbation in water vapour", yielding a negative proportionality between perturbations in the specific volume and those in the water vapour density ($\alpha' \propto -\rho_v'$). Note that the context of this relationship is the Webb et al. (1980) paper, which assumes constant pressure. Excluding temperature effects (the other WPL correction), the effect of isobaric and isothermal evaporation is to humidify the air. According to the ideal gas law (equation 4 of Webb et al. (1980)), under such conditions the total number of molecules per unit volume remains constant, such that – for a fixed Eulerian volume – dry air molecules disappear at the same rate that water vapour molecules appear due to humidification. Since water vapour has less mass than does dry air, the effect of such humidification is a reduction in air density, and hence an increase in the specific volume, demonstrating that the proportionality between perturbations in the specific volume and those in the water vapour density is in fact positive. Therefore, I believe that equation C2 of the Paw U paper is patently incorrect. Since the entirety of Section 3 based on this, I recommend its elimination.

**RESPONSE** *I agree that Paw U et al.'s statement that $\alpha' \propto -\rho_v'$ violates physical reality. This can be shown relatively simply. By definition $\alpha = 1/\rho_a$, from which is immediately follows that $\alpha' = -\rho_a'/\bar{\rho}_a^2$. Next using the displacement assumption, $\rho_v' + \rho_a' = 0$, it also follows that $\rho_v' = -\rho_a'$. Therefore, $\alpha' = \rho_v'/\bar{\rho}_a^2$ or $\alpha' \propto \rho_v'$, which is now in concordance with the ASK's analysis. For my restatement of Paw U et al.'s*

*derivation of $T_e'$ and $H_e'$ (their Appendix C) I simply maintained the correct sign and focused on what Paw U et al. would have derived if they had not made a sign error. As I said in an earlier response I am less concerned about specific errors than I am in trying to understand the physics of the idea that the authors are trying to express.*

**ASK - Page 6, line 24:** This part of the paper (if not eliminated) would be more clear if $T_e'$ were defined more explicitly, as "the temperature perturbation equivalent to the energy needed for expansion" as in the Paw U paper.

**RESPONSE** *I concur. The manuscript has been changed.*

---

## Referee Comment (RC3) · Grant Petty (Referee) · 15 Aug 2019

The topic addressed by this paper is surprisingly subtle and complex, which is probably why it has not already been definitively addressed in the past. Bill Massman offers what appears to be a rather rigorous analysis that seems plausible on its face, and it certainly leads to the conclusions one might expect, which is that the effects of second-order corrections are far too small to account for the commonly reported closure problem in surface energy budget studies.

However, even after multiple readings, I have still not completely convinced myself that there couldn't be an error or inconsistency in assumptions buried somewhere in the

analysis that affects the precise conclusions. I recommend publication anyway with the thought that (a) it may well be correct, and (b), even if not, it will at least provide a useful starting point for others who may wish to reexamine this problem in the future.

A number of specific issues have already been addressed by other reviewers, and Dr. Massman has already responded to many of those. Here I focus only on the things that caught my attention as I was reviewing the manuscript:

1) With regard to this paper's reference to the Kowalski note, it's not completely clear to me that Massman's section 2 is even really examining the same physical issue. In particular, line 20 on p. 2 states, "The purpose of the present paper is to examine the methods and conclusions of these two papers." But Massman doesn't actually examine Kowalski's *methods*, as far as I can tell. And Massman is looking at the role of non-ideality, whereas my recollection of Kowalski's contribution (which was withdrawn) was that it was looking at a possibly missing contribution of $pV$ work in the enthalpy of evaporation (I don't have the link to the Kowalski manuscript at my fingertips so can't verify). In any case, if Kowalski's unpublished (except as a discussion paper) work is referenced at all – and I'm not necessarily sure it should be, I think the physical and logical relationship between the problems Massman and Kowalski were considering (irrespective of the methods employed) should be made more explicit.

2) There do seem to be some potential inconsistencies in assumptions. These may not be fatal, but the author should perhaps acknowledge them and explain why they don't undermine some of the conclusions. For example:

a) In lines 20–25 of p. 3, the system is considered to be isolated, including no mechanical interaction with the environment. By definition, this implies constant volume, yet line 4 of p. 5 states, "The final step is to specify whether the enthalpic change occurs at a constant pressure or at a constant volume." The reality, of course, is that pressure is normally very close to hydrostatic in the boundary layer, so a constant pressure assumption seems more germaine. In fact, if I were attempting the analysis myself, I

might consider evaporation in a constant volume system as an intermediate stage for analytical convenience, with subsequent adiabatic expansion to the ambient pressure.

b) Section 2 is explicitly looking at the effects of non-ideality, but line 31 on p. 3 states that $p_a = p_d + p_{v,sat}$, implying that Dalton's law of partial pressures is exact for this system. Doesn't the existence of non-zero $B_a$ (2nd virial coeff. for moist air) imply that the final pressure will be greater or less than the sum of the individual pressures?

3) I would have liked to see more slightly more context for equation (5) at the bottom of p. 4. For those who don't normally work with the virial coefficients, what does the equation of state look like when $B_a$, $B_d$, and $B_v$ are included, and how does (5) arise from that equation and from the definition of $I_B/\chi_v$?

4) line 12, p. 5: Under what conditions might evaporation occurring on the Earth's surface be poorly approximated as isobaric? I can't think of any, except perhaps in the interior of a leaf with very high stomatal resisistance, and I'm not even completely persuaded in that case.

---

## Author Comment (AC3) · 6 Sep 2019

**Response to Comments from Grant Petty dated 15 August 2019**

My thanks to Grant Petty for his comments. They were helpful. My response (in italics) follow a repeat of his comment.

**GP (0-a)** The topic addressed by this paper is surprisingly subtle and complex, which is probably why it has not already been definitively addressed in the past. Bill Massman offers what appears to be a rather rigorous analysis that seems plausible

on its face, and it certainly leads to the conclusions one might expect, which is that the effects of second-order corrections are far too small to account for the commonly reported closure problem in surface energy budget studies.

**RESPONSE (0-a)** *I agree.*

**GP (0-b)** However, even after multiple readings, I have still not completely convinced myself that there couldn't be an error or inconsistency in assumptions buried somewhere in the analysis that affects the precise conclusions. I recommend publication anyway with the thought that (a) it may well be correct, and (b), even if not, it will at least provide a useful starting point for others who may wish to reexamine this problem in the future.

**RESPONSE (0-b)** *Thank you and I concur.*

**GP (0-c)** A number of specific issues have already been addressed by other reviewers, and Dr. Massman has already responded to many of those. Here I focus only on the things that caught my attention as I was reviewing the manuscript:

**GP 1)** With regard to this paper's reference to the Kowalski note, it's not completely clear to me that Massman's section 2 is even really examining the same physical issue. In particular, line 20 on p. 2 states, "The purpose of the present paper is to examine the methods and conclusions of these two papers." But Massman doesn't actually examine Kowalski's methods, as far as I can tell. And Massman is looking at the role of non-ideality, whereas my recollection of Kowalski's contribution (which was withdrawn) was that it was looking at a possibly missing contribution of $pV$ work in the enthalpy of evaporation (I don't have the link to the Kowalski manuscript at my fingertips so can't verify). In any case, if Kowalski's unpublished (except as a discussion paper) work is referenced at all – and I'm not necessarily sure it should be, I think the physical and logical relationship between the problems Massman and

Kowalski were considering (irrespective of the methods employed) should be made more explicit.

**RESPONSE 1)** *I have revised the paragraph to eliminate the suggestion that I am examining Kowalski's methods. The paragraph now reads:*
***The present paper employs "classical" thermodynamics to examine (a) the influence that the non-ideality of atmospheric gases can have on the SEB and (b) the methods and conclusions of Paw U et al. (2000, Appendix C) regarding the first law of thermodynamics and the pressure work term's influence on the turbulent heat flux and ultimately the SEB as well. Although it is true that what I develop herein is not necessarily "new" science, some of the theory I employ may well be new to the general environmental and geo-biophysical communities. The present study is divided into two parts. The first examines and quantifies how mixing of air and water vapor as non-ideal (or real) gases, rather than as ideal gases, can have on $L_v$ and the specific heat of moist air. In the second part the first law of thermodynamics is employed to derive the influence water vapor has on potential temperature, which in turn gives rise to an expression, different from that developed by Paw U et al. (2000, Appendix C), relating how the kinematic heat flux is influenced by the mass flux of water vapor, $E$. In summary, this study shows that any potential corrections to the SEB from either of these two sources are likely to be negligible and certainly much smaller than either Kowalski (2018) or Paw U et al. (2000) propose.***

**GP 2)** There do seem to be some potential inconsistencies in assumptions. These may not be fatal, but the author should perhaps acknowledge them and explain why they don't undermine some of the conclusions. For example:

**GP 2a)** In lines 20–25 of p. 3, the system is considered to be isolated, including no mechanical interaction with the environment. By definition, this implies constant

volume, yet line 4 of p. 5 states, "The final step is to specify whether the enthalpic change occurs at a constant pressure or at a constant volume." The reality, of course, is that pressure is normally very close to hydrostatic in the boundary layer, so a constant pressure assumption seems more germane. In fact, if I were attempting the analysis myself, I might consider evaporation in a constant volume system as an intermediate stage for analytical convenience, with subsequent adiabatic expansion to the ambient pressure.

**RESPONSE 2a)** *I have revised the manuscript so that the paragraph above Equation (3) now reads as below. This revision also eliminates Kowalski's concern about the flow of the manuscript being interrupted by the interjection of "Note ..." about the enhancement factor $f$ (page 4, lines 12-16 of the original manuscript).*
***For an ideal gas the final pressure is $p_{v,sat}$ (Pa), but for a non-ideal gas the saturated vapor pressure is $fp_{v,sat}$ (Hyland and Wexler 1983; Goff 1949), where $f = f(T_K, p_a)$ is termed the enhancement factor and $1 < f < 1.006$ near STP (Hyland and Wexler 1983; Nelson and Sauer 2004). Consequently, the final pressure of the water vapor will exceed $p_{v,sat}$ by a small amount. On the other hand, the final pressure of the dry air, $p_{d,final}$ (Pa), will be slightly less that $p_d$ because the final gas volume of the system will be slightly greater than the initial volume due to the decrease in the volume of liquid with the evaporative loss of $N_v$ mols of liquid. In the present scenario this difference between the final and initial pressures is small: $\approx 0.001 p_d$. Because both $f$ and this relative pressure difference are so small and they tend to compensate for one another, it is reasonable to ignore both effects and approximate the final total pressure, $p_a$ (Pa), as simply as $p_a = p_d + p_{v,sat}$; meaning that the present purposes evaporation occurring within an isolated system can be considered as an archetypical constant pressure process. Nonetheless, it is also worth emphasizing that, in fact, evaporation in the present isolated system (as well as within the atmospheric surface layer) is neither a constant volume, nor a constant pressure, process. Rather it is a combination or hybrid of the two***

***processes.***

**GP 2b)** Section 2 is explicitly looking at the effects of non-ideality, but line 31 on p. 3 states that $p_a = p_d + p_{v,sat}$, implying that Dalton's law of partial pressures is exact for this system. Doesn't the existence of non-zero $B_a$ (2nd virial coeff. for moist air) imply that the final pressure will be greater or less than the sum of the individual pressures?

**RESPONSE 2b)** *The reviewer is correct. Non-ideality does imply that the final pressure is different than for an ideal case, for which Dalton's law implies $p_a = p_d + p_{v,sat}$. The revisions – see my RESPONSE 2a) above – now address this issue plus another issue related to the final pressure of the dry air. I think my revisions should cover Dr. Petty's concern.*

**GP 3)** I would have liked to see more slightly more context for equation (5) at the bottom of p. 4. For those who don't normally work with the virial coefficients, what does the equation of state look like when $B_a$, $B_d$, and $B_v$ are included, and how does (5) arise from that equation and from the definition of $I_B/\chi_v$? **RESPONSE 3)** *I have revised the manuscript to accommodate GP's request. The text now reads:*
***In general $I_B$ is expressed in terms of the second and third virial coefficients (Hyland and Wexler 1983; Wagner and Pruß 2002), which are defined by the virial equation of state (Hyland and Wexler 1983; Sattar 2000) as follows:***

$$\frac{p_i v_i}{R T_K} = 1 + \frac{B_i}{v_i} + \frac{C_i}{v_i^2} + \cdots \quad (6)$$

***where the subscript 'i' refers to water vapor ($i = v$), dry air ($i = d$), or moist air ($i = a$); $B_i$ (m$^3$ mol$^{-1}$) is the second virial coefficient, $C_i$ (m$^6$ mol$^{-2}$) is the third virial coefficient, and in general $B_i$ and $C_i$ are both functions of temperature, $T_K$; $p_i$ is the gas pressure (Pa) and $v_i$ is the molar volume (m$^3$ mol$^{-1}$) of the gas. For this study it is sufficient to consider only the second virial coefficients. For***

**dry air and water vapor,** $B_i = B_i(T_K)$ **is determined by empirical curve fitting of observed data. For this study** $B_v(T_K)$ **is taken from Equation (6) of Harvey and Lemon (2004) and** $B_d(T_K)$ **is taken from Equation (10) of Hyland and Wexler (1983). Because moist air is a mixture of dry air and water vapor the second virial coefficient for moist air takes the form** $B_a = \chi_v^2 B_v + 2\chi_v\chi_d B_{vd} + \chi_d^2 B_d$ **(Sattar 2000), where** $B_{vd}$ **(m$^3$ mol$^{-1}$) is the cross virial coefficient for moist air. For the present study** $B_{vd}(T_K)$ **is taken from Equation (15) of Hyland and Wexler (1983). Once the equation of state has been specified, the general expression for** $I_B$ **can be derived (e.g., Sattar 2000), yielding . . .**

*But note, I have not elaborated on the methods used for employing the virial equation of state to produce $I_B$ or equivalently $I_B/\chi_v$ = Equation (7) of the revised text. I just simply refer to Sattar (2000), who only briefly sketches what is needed for this step. This is a non-trivial step and needs some understanding of the kinetic theory of gases and the electrostatic interaction potential (e.g., Lennard-Jones potential) between water molecules and other water molecules and between water molecules and dry air molecules, which I think is well beyond the intent of the present study. But knowing that $H_2O$ is a polar molecule and that $O_2$ and $N_2$ are non-polar is sufficient to recognize that the major component of $B_a$ is $B_v$, followed by $B_{vd}$ and that $B_d$ is essentially negligible.*

**GP 4)** line 12, p. 5: Under what conditions might evaporation occurring on the Earth's surface be poorly approximated as isobaric? I can't think of any, except perhaps in the interior of a leaf with very high stomatal resistance, and I'm not even completely persuaded in that case.

**RESPONSE 4)** *I agree that it is more logical to assume that evaporation into the atmosphere occurs at constant pressure than at constant volume. But as I pointed out in the revision discussed under 2a) above evaporation occurs by a process that is neither constant pressure nor constant volume, but is really a mixture of the two*

*pathways. In which case, it is important to consider both pathways since the results will bound the true enthalpy of vaporization and the specific heat estimates derived in this study. More importantly though it should not be surprising that these two pathways yield nearly identical results, except maybe under extremely hot and moist conditions (figures 2 and 3). Nonetheless, I added a sentence to the paragraph that follows Equation (7) of the revised text to suggest that evaporation into the atmosphere is better approximated by a constant pressure pathway than a constant volume pathway. These two sentences now read:*

**The final step is to specify whether the enthalpic change occurs at constant pressure or at constant volume. Although assuming a constant pressure pathway for modeling evaporation into the atmosphere is likely to be more appropriate than assuming a constant volume pathway, both pathways need to be considered here because any evaporation occurring on the earth's surface is going to lie somewhere between these two (bounding) pathways.**

---

## Author Response (AR1)

**Author's Summary Response and Guide to Manuscript Changes - 23 October 2019**

My thanks to the reviewers and editors for their efforts. The following is a summary of the changes I have made to the manuscript. The revisions required an extensive expansion and rewriting of the material and to correct several errors I made in the previous draft. Here I highlight the major changes to the most critical issues raised by the editors and the reviewers.

**Lines 29-36:** Explanation of the error made by Kowalski (2018) – editor's comment 07-10-2019. The error made by Kowalski (2018) is now clearly articulated. Furthermore, Kowalski (2018) is cited in the present paper because it very germane to the subject, methods and intent of my paper. After reading Kowalski (2018) and the comments I thought that the issues being addressed by Kowalski (2018) and Paw U et al. (2004) needed some discussion and highlighting in the literature, so my paper cites and comments on Kowalski (2018). I did not articulate in detail why I cite Kowalski (2018) in the present manuscript; but I think this is largely obvious and I don't think these details benefit either the paper or the reader. But I no longer say that I am "examining the methods of Kowalski (2018)" in the present version of the manuscript.

**Lines 44-54:** Restating and clarifying the purpose of my paper. I eliminated the statement that I am "examining the methods of Kowalski (2018)" in the present version of my manuscript – in response to comments from Petty and the editor.

**Paragraph, Lines 75-84:** Quantification and explanation of $du^*$, $p^*\Delta v^*$ and the approximation to $p^*\Delta v^*$ in Figure 1 – suggested by the anonymous editor. Also further serves my purpose of highlighting $p\Delta v$ in this paper.

**Lines 85-124:** A more precise calculation of the change in enthalpy associated with evaporation. This revision corrects the error in the previous version of the manuscript that was pointed out by Kowalski.

**Lines 129-140:** Definition/discussion of the virial equation of state – as requested by Petty.

**Lines 142-145:** Assertion/clarification that the process of evaporation into the free atmosphere lies between a constant pressure process and constant volume process – in response to Petty.

**Lines 167-182:** Correction to the calculation for $\Delta C_p$. My original expression equating $C_p$ to $dL_v/dT$ was in error. Revision in response to comment by Kowalski.

**Lines 211-226:** Explanation of the error(s) made by Paw U et al. (2004) – editor's comment 07-10-2019. My original impression of the mistake Paw U et al's. (2004) made was that it was largely a simple sign error. But upon closer examination, I realized that the error was more subtle than I originally thought. Consequently, the present revision/clarification required an additionally paragraph, as well as, some speculation on my part about why/how Paw U et al. may have made the error in the first place.

**Lines 302-304:** Code and data availability The manuscript now provides a doi number for accessing the computer code along with a table of key numerical outputs produced with that code.

**Appendix A, Lines 306-337:** More precise calculation of the change in temperature of the system, $\delta T$, due to the evaporation of water from the water reservoir – in response to concerns expressed by the editor and Kowalski.

**Figure 1.** I have revised this figure. It now includes labeled axes on both the left and right sides of the figure – as requested by the anonymous editor.

**Figure 3.** I have revised this figure to include the correction to and recalculation of $\Delta C_p / C_p$ – in response to Kowalski.

---

## Referee Report (RR1)

**Review of Revised Version of HESS Manuscript #2019-153**

Title:   Impacts of non-ideality and the thermodynamic pressure work term p_v on the Surface Energy Balance

Author: Massman

**Review**

The author has responded appropriately.

In particular, Fig. 1 (and associated text) now has a more complete explanation of internal energy and enthalpy.

Publish it!

---

## Referee Report (RR2)

Review of HESS Manuscript #2019-153, revised version
Title: Impacts of non-ideality and the thermodynamic pressure work term pΔv on the Surface Energy Balance
Author:Massman
Review by A. S. Kowalski

This revised manuscript is much improved in comparison with the initial submission, and I suggest that it be published subject to minor revisions. I have tried to organize my recommendations in order of importance.

1. Line 245: The author claims to identify an error in equation (2.66) of Curry and Webster (1999), but I see no error in that equation. The coefficient 0.2 modifying $q_v$ derives from a binomial expansion and approximation, multiplying both numerator and denominator by the same factor (1 - 0.*87 $q_v$*) and then neglecting quadratic terms to simplify the result (since $q_v^2$ << 1). This can be found in other texts as well (e.g., Rogers and Yau, 1988, A Short Course in Cloud Physics, Pergamon, Oxford). In equation (15) and all subsequent equations that contain the factor 0.33, I believe that this should be changed back to the coefficient 0.2. This may also change the percentage that appears in the conclusions (line 298).

2. Line 284: The "displacement assumption" of Pau U et al. (2000) can hardly be brought into question, since it falls directly out of the Ideal Gas Law for the conditions that they assumed. The context of the Paw U et al. paper is evaporation that is both isobaric (as assumed by the Webb et al. paper under consideration) and isothermal (excluding temperature effects – i.e., the WPL vapour correction). In such a context, equation (4) of Webb et al. (1980) is a version of the Ideal Gas Law that adequately justifies the relevance of water vapour displacing dry air. I appreciate the author's argument that evaporation is truly neither a constant volume, nor a constant pressure process (line 103), but I do not think that it justifies the wording used here.

3. Lines 230-240: There is something inconsistent about beginning an argument for defining the heat flux using potential temperature (rather than the temperature) with an equation that is valid only for an *incompressible* atmospheric process. It may be preferable to use the proper definition of the material derivative as $\frac{d\theta}{dt} = \frac{\partial\theta}{\partial t} + \boldsymbol{u} \cdot \boldsymbol{\nabla}\theta$ and so be able to remove the word "incompressible". Perhaps even simpler would be to simply state that the potential temperature is the key variable for discussion, and cite an appropriate reference (e.g., Kowalski, A. S. and Argüeso, D., 2011, *Tellus*, **63B**, 1059-1066).

4. I find the author's use of temperature ranges to be inconsistent and frankly inexplicable, resulting in an unnecessary distraction from the message of the paper. Line 57 suggests examination of the surface energy budget near STP (i.e., not far from 0ºC), which seems appropriate if somewhat vague. However, temperature ranges are later defined variously throughout the manuscript (all converted to ºC here) as:
   a. Line 59: 0 – 100ºC;
   b. Line 155: 3 – 42ºC;
   c. Line 185: 7 – 77ºC; and

     d.  Line 298: 12 – 52ºC.

   I believe that a more appropriate range of "temperatures commonly encountered with micrometeorological techniques" (line 156) would be something like -35º - 45ºC. If the "extrapolation" of Dr. Massman's results to such a range in any way changes his calculations, then some revision may be required that might not classify as "minor".

5. The use of both mass- and molar-based definitions of the specific heat is similarly distracting. I see little point in defining the molar specific heat when its use complicates the "final result" (as in equation 13 which, if I am not mistaken, has disguised the mass specific heat as the ratio $c_v/\mu$).
6. Throughout the manuscript, units are specified with no space separating them. So for example at line 16, I think that "Wm$^{-2}$" should be changed to "W m$^{-2}$", and likewise in many subsequent instances. This is particularly egregious at line 43, where the characters "kgm" appear in succession.
7. At line 69, delete the first instance of the word "pure".
8. At line 72, "unnecessary **to** consider"
9. At line 75, "components of the **specific** enthalpy"

I hope that some fraction of these suggestions will be helpful when producing the final version of the manuscript.

---

## Author Response (AR2)

**Response to Comments from reviewers dated (approximately) 10 January 2020**
**and**
**Line-by-Line Summary of changes to the manuscript**

**RESPONSE**

My thanks to all reviewers for their comments. They were helpful. In response to Dr. Petty's comments I spell checked the manuscript and made all necessary (three) corrections. The Anonymous Referee #1 is satisfied with my changes. No further response is necessary to either of these two reviewers. Dr. Kowlaski (ASK) suggested several minor revisions. My response to ASK follows.

**ASK:** This revised manuscript is much improved in comparison with the initial submission, and I suggest that it be published subject to minor revisions. I have tried to organize my recommendations in order of importance. **RESPONSE:** *OK.*

**ASK (1.)** Line 245: The author claims to identify an error in equation (2.66) of Curry and Webster (1999), but I see no error in that equation. The coefficient 0.2 modifying $q_v$ derives from a binomial expansion and approximation, multiplying both numerator and denominator by the same factor (1 - 0.87$q_v$) and then neglecting quadratic terms to simplify the result (since $q_v^2 \ll 1$). This can be found in other texts as well (e.g., Rogers and Yau, 1988, A Short Course in Cloud Physics, Pergamon, Oxford). In equation (15) and all subsequent equations that contain the factor 0.33, I believe that this should be changed back to the coefficient 0.2. This may also change the percentage that appears in the conclusions (line 298).
**RESPONSE:** *I agree. I misunderstood $c_{pv}$ in Equation (2.66) of Curry and Webster (1999). Sorry for the confusion. All necessary corrections were made and the sentence pointing out the putative error has been removed.*

**ASK (2.)** Line 284: The 'displacement assumption' of Paw U et al. (2000) can hardly be brought into question, since it falls directly out of the Ideal Gas Law for the conditions that they assumed. The context of the Paw U et al. paper is evaporation that is both isobaric (as assumed by the Webb et al. paper under consideration) and isothermal (excluding temperature effects, i.e., the WPL vapour correction). In such a context, equation (4) of Webb et al. (1980) is a version of the Ideal Gas Law that adequately justifies the relevance of water vapour displacing dry air. I appreciate the author's argument that evaporation is truly neither a constant volume, nor a constant pressure process (line 103), but I do not think that it justifies the wording used here.
**RESPONSE:** *No Change. I disagree. Nowhere in the derivation of the WPL terms, whether by an isobaric or isothermal process or both, is the assumption made that an evaporating molecule (mol) of water vapor exactly replaces a molecule (mol) of dry air. Yet that is precisely what Paw U et al. do (i.e., $\varrho_a' = -\varrho_v'$). Furthermore, (1) I don't think Paw U et al. would necessarily agree that their derivation assumes that evaporation is an isothermal process (it clearly is not: see my Appendix) and (2), and I think the reviewer would agree with me, that it makes no physical sense to assume that the process of evaporation is an isothermal isobaric process. Finally, I am not justifying my wording on the basis of my assertion that evaporation lies somewhere between a constant volume and a constant pressure process. I justify my statement on the fact that the result proposed by Paw U et al. did not give a result that is consistent with*

*what I obtained using an Equation (15), which basically is an equation of state for a moist atmosphere. It seems to me that Paw U et al.'s displacement assumption leads to a contradiction and specifically contradicts what one might expect from the appropriate equation of state for moist air.*

**ASK (3.)** Lines 230-240: There is something inconsistent about beginning an argument for defining the heat flux using potential temperature (rather than the temperature) with an equation that is valid only for an incompressible atmospheric process. It may be preferable to use the proper definition of the material derivative as $d\theta/dt = \partial\theta/\partial t + u \bullet \nabla\theta$ and so be able to remove the word 'incompressible'. Perhaps even simpler would be to simply state that the potential temperature is the key variable for discussion, and cite an appropriate reference (e.g., Kowalski, A. S. and Argueso, D., 2011, Tellus, 63B, 1059-1066).
**RESPONSE:** *Fair point. The following changes were made: Equation (14) now includes the term associated with compressibility and I have removed the word 'incompressible' from the preceding sentence. I have also added the following sentence: "Here the term associated with compressible effects, $\theta\nabla \bullet \boldsymbol{u}$, is included for completeness, although it is not important for the present purposes."*

**ASK (4.)** I find the author's use of temperature ranges to be inconsistent and frankly inexplicable, resulting in an unnecessary distraction from the message of the paper. Line 57 suggests examination of the surface energy budget near STP (i.e., not far from 0C), which seems appropriate if somewhat vague.
**RESPONSE:** *The sentence following one that the reviewer cites states 'Here "near STP" will be understood as pressures between about 70 kPa and 105 kPa and temperatures between about 0 C and 100 C or so – or an atmospheric state typical of near-surface conditions on earth.' I am not sure why the reviewer finds my use of the words "near STP" vague.*
**CONTINUING ASK (4.)** However, temperature ranges are later defined variously throughout the manuscript (all converted to C here) as:
a. Line 59: 0 - 100C; **RESPONSE:** *At this point of the text this range of temperatures is appropriate since all calculations are done between these temperatures and all graphs cover this range of temperatures.*
b. Line 155: 3 - 42C; **RESPONSE:** *Changed to 3 - 52C*
c. Line 185: 7 - 77C; *Changed to 3 - 52C* and
d. Line 298: 12 - 52C. **RESPONSE:** *Changed to 3 - 52C*
**CONTINUING ASK (4.)** I believe that a more appropriate range of temperatures commonly encountered with micrometeorological techniques (line 156) would be something like -35 to 45C. If the extrapolation of Dr. Massman's results to such a range in any way changes his calculations, then some revision may be required that might not classify as minor.
**RESPONSE:** *The following change was made: The sentence on line 156 now reads 'With the exception of sublimation of ice or snow, these results suggests that surface energy fluxes associated with ET measured at temperatures commonly encountered with micrometeorological techniques (i.e., between about 275 and 325 K)...'; where the underlined portion of the text has been added to the original text.*
**ADDITIONAL RESPONSE:** *The reviewer is correct I did am not considering sub-zero C temperatures. But it is interesting that the energy balance closure tends to be worse over ice and snow. I am left to wonder now if it might be worth investigating the influence of the thermodynamics of sublimation on the surface energy balance.*

**ASK (5.)** The use of both mass- and molar-based definitions of the specific heat is similarly distracting. I see little point in defining the molar specific heat when its use complicates the 'final result' (as in equation 13 which, if I am not mistaken, has disguised the mass specific heat as the ratio $c_v/\mu$).
**RESPONSE:** *No Change. The reviewer is correct about $\mu$ being buried in the final result. That is because I wanted my final result to be directly comparable to the expression developed by Paw U et al. (2000) and that takes careful management of the units. It is also important that the end result be consistent with my reworking of Paw U et al.'s derivation and their mol-per-mol displacement assumption, which is in molar units. Sorry if this causes confusion; but, switching units does not concern me that much because the conversion is linear and therefore straightforward.*

**ASK (6.)** Throughout the manuscript, units are specified with no space separating them. So for example at line 16, I think that 'Wm$^{-2}$' should be changed to 'W m$^{-2}$', and likewise in many subsequent instances. This is particularly egregious at line 43, where the characters 'kgm' appear in succession.
**RESPONSE:** *This seems to be a journal formatting issue. I am not sure how to change this.*

**ASK (7.)** At line 69, delete the first instance of the word 'pure'. ***The manuscript has been changed.***

**ASK (8.)** At line 72, 'unnecessary to consider' ***The manuscript has been changed.***

**ASK (9.)** At line 75, 'components of the specific enthalpy' ***The manuscript has been changed.***

I hope that some fraction of these suggestions will be helpful when producing the final version of the manuscript. ***They were.***

**SUMMARY**

The following lists the line numbers of the revised manuscript where changes were made to the manuscript.

lines 30, 32, 69, 72, 75 – Minor editorial (spelling and grammatical) changes – **All Reviewers**
lines 154-156; line 186 – **ASK (4.)**
lines 231-236 – **ASK (3.)**
lines 243-245; line 252; lines 256-259; lines 265-266; line 272; line 284 – **ASK (1.)**
lines 299-301 – **ASK (1.)** & **ASK (4.)**